# A BETTER ALTERNATIVE TO ERROR FEEDBACK FOR COMMUNICATION-EFFICIENT DISTRIBUTED LEARNING

**Samuel Horváth and Peter Richtárik**
KAUST
Thuwal, Saudi Arabia
{samuel.horvath, peter.richtarik}@kaust.edu.sa

## ABSTRACT

Modern large-scale machine learning applications require stochastic optimization algorithms to be implemented on distributed compute systems. A key bottleneck of such systems is the communication overhead for exchanging information (e.g., stochastic gradients) across the workers. Among the many techniques proposed to remedy this issue, one of the most successful is the framework of compressed communication with error feedback (EF). EF remains the only known technique that can deal with the error induced by contractive compressors which are not unbiased, such as Top-$K$ or PowerSGD. In this paper, we propose a new and theoretically and practically better alternative to EF for dealing with contractive compressors. In particular, we propose a construction which can transform any contractive compressor into an induced unbiased compressor. Following this transformation, existing methods able to work with unbiased compressors can be applied. We show that our approach leads to vast improvements over EF, including reduced memory requirements, better communication complexity guarantees and fewer assumptions. We further extend our results to federated learning with partial participation following an arbitrary distribution over the nodes, and demonstrate the benefits thereof. We perform several numerical experiments which validate our theoretical findings.

## 1 INTRODUCTION

We consider distributed optimization problems of the form

$$\min_{x \in \mathbb{R}^d} f(x) \coloneqq \tfrac{1}{n} \sum_{i=1}^{n} f_i(x), \tag{1}$$

where $x \in \mathbb{R}^d$ represents the weights of a statistical model we wish to train, $n$ is the number of nodes, and $f_i \colon \mathbb{R}^d \to \mathbb{R}$ is a smooth differentiable loss function composed of data stored on worker $i$. In a classical distributed machine learning scenario, $f_i(x) \coloneqq \mathrm{E}_{\zeta \sim \mathcal{D}_i}\left[f_\zeta(x)\right]$ is the expected loss of model $x$ with respect to the local data distribution $\mathcal{D}_i$ of the form, and $f_\zeta \colon \mathbb{R}^d \to \mathbb{R}$ is the loss on the single data point $\zeta$. This definition allows for different distributions $\mathcal{D}_1, \ldots, \mathcal{D}_n$ on each node, which means that the functions $f_1, \ldots, f_n$ can have different minimizers. This framework covers Stochastic Optimization when either $n = 1$ or all $\mathcal{D}_i$ are identical, Empirical Risk Minimization (ERM), when $f_i(x)$ can be expressed as a finite average, i.e, $f_i(x) = \frac{1}{m_i} \sum_{i=1}^{m_i} f_{ij}(x)$ for some $f_{ij} \colon \mathbb{R}^d \to \mathbb{R}$, and Federated Learning (FL) (Kairouz et al., 2019) where each node represents a client.

**Communication Bottleneck.** In distributed training, model updates (or gradient vectors) have to be exchanged in each iteration. Due to the size of the communicated messages for commonly considered deep models (Alistarh et al., 2016), this represents significant bottleneck of the whole optimization procedure. To reduce the amount of data that has to be transmitted, several strategies were proposed.

One of the most popular strategies is to incorporate local steps and communicated updates every few iterations only (Stich, 2019a; Lin et al., 2018a; Stich & Karimireddy, 2020; Karimireddy et al.,

2019a; Khaled et al., 2020). Unfortunately, despite their practical success, local methods are poorly understood and their theoretical foundations are currently lacking. Almost all existing error guarantees are dominated by a simple baseline, minibatch SGD (Woodworth et al., 2020).

In this work, we focus on another popular approach: *gradient compression*. In this approach, instead of transmitting the full dimensional (gradient) vector $g \in \mathbb{R}^d$, one transmits a compressed vector $\mathcal{C}(g)$, where $\mathcal{C} : \mathbb{R}^d \to \mathbb{R}^d$ is a (possibly random) operator chosen such that $\mathcal{C}(g)$ can be represented using fewer bits, for instance by using limited bit representation (quantization) or by enforcing sparsity. A particularly popular class of quantization operators is based on random dithering (Goodall, 1951; Roberts, 1962); see (Alistarh et al., 2016; Wen et al., 2017; Zhang et al., 2017; Horváth et al., 2019a; Ramezani-Kebrya et al., 2019). Much sparser vectors can be obtained by random sparsification techniques that randomly mask the input vectors and only preserve a constant number of coordinates (Wangni et al., 2018; Konečný & Richtárik, 2018; Stich et al., 2018; Mishchenko et al., 2019b; Vogels et al., 2019). There is also a line of work (Horváth et al., 2019a; Basu et al., 2019) in which a combination of sparsification and quantization was proposed to obtain a more aggressive effect. We will not further distinguish between sparsification and quantization approaches, and refer to all of them as compression operators hereafter.

Considering both practice and theory, compression operators can be split into two groups: biased and unbiased. For the unbiased compressors, $\mathcal{C}(g)$ is required to be an unbiased estimator of the update $g$. Once this requirement is lifted, extra tricks are necessary for Distributed Compressed Stochastic Gradient Descent (DCSGD) (Alistarh et al., 2016; 2018; Khirirat et al., 2018) employing such a compressor to work, even if the full gradient is computed by each node. Indeed, the naive approach can lead to exponential divergence (Beznosikov et al., 2020), and Error Feedback (EF) (Seide et al., 2014; Karimireddy et al., 2019b) is the only known mechanism able to remedy the situation.

**Contributions.** Our contributions can be summarized as follows:

• **Induced Compressor.** When used within the stabilizing EF framework, biased compressors (e.g., Top-$K$) can often achieve superior performance when compared to their unbiased counterparts (e.g., Rand-$K$). This is often attributed to their low variance. However, despite ample research in this area, EF remains the only known mechanism that allows the use of these powerful biased compressors. Our key contribution is the development of a simple but remarkably effective alternative—and this is the only alternative we know of—which we argue leads to better and more versatile methods both in theory and practice. In particular, we propose a general construction that can transform any biased compressor, such as Top-$K$, into an unbiased one for which we coin the name *induced compressor* (Section 3). Instead of using the desired biased compressor within EF, our proposal is to instead use the induced compressor within an appropriately chosen existing method designed for unbiased compressors, such as distributed compressed SGD (DCSGD) (Khirirat et al., 2018), variance reduced DCSGD (DIANA) (Mishchenko et al., 2019a) or accelerated DIANA (ADIANA) (Li et al., 2020). While EF can bee seen as a version of DCSGD which can work with biased compressors, variance reduced nor accelerated variants of EF were not known at the time of writing this paper.

• **Better Theory for DCSGD.** As a secondary contribution, we provide a new and tighter theoretical analysis of DCSGD under weaker assumptions. If $f$ is $\mu$-quasi convex (not necessarily convex) and local functions $f_i$ are $(L, \sigma^2)$-smooth (weaker version of $L$-smoothness with strong growth condition), we obtain the rate $\mathcal{O}\left(\delta_n L r^0 \exp\left[-\frac{\mu T}{4\delta_n L}\right] + \frac{(\delta_n - 1)D + \delta\sigma^2/n}{\mu T}\right)$, where $\delta_n = 1 + \frac{\delta - 1}{n}$ and $\delta \geq 1$ is the parameter which bounds the second moment of the compression operator, and $T$ is the number of iterations. This rate has linearly decreasing dependence on the number of nodes $n$, which is strictly better than the best-known rate for DCSGD with EF, whose convergence does not improve as the number of nodes increases, which is one of the main disadvantages of using EF. Moreover, EF requires extra assumptions. In addition, while the best-known rates for EF (Karimireddy et al., 2019b; Beznosikov et al., 2020) are expressed in terms of functional values, our theory guarantees convergence in both iterates and functional values. Another practical implication of our findings is the reduction of the memory requirements by half; this is because in DCSGD one does not need to store the error vector.

• **Partial Participation.** We further extend our results to obtain the first convergence guarantee for partial participation with arbitrary distributions over nodes, which plays a key role in Federated Learning (FL).

**Algorithm 1** DCSGD

1: **Input:** $\{\eta^k\}_{k=0}^T > 0$, $x_0$
2: **for** $k = 0, 1, \ldots T$ **do**
3:    **Parallel: Worker side**
4:    **for** $i = 1, \ldots, n$ **do**
5:      obtain $g_i^k$
6:      send $\Delta_i^k = \mathcal{C}^k(g_i^k)$ to master
7:      [no need to keep track of errors]
8:    **end for**
9:    **Master side**
10:   aggregate $\Delta^k = \frac{1}{n} \sum_{i=1}^n \Delta_i^k$
11:   broadcast $\Delta^k$ to each worker
12:   **Parallel: Worker side**
13:   **for** $i = 1, \ldots, n$ **do**
14:     $x^{k+1} = x^k - \eta^k \Delta^k$
15:   **end for**
16: **end for**

**Algorithm 2** DCSGD with Error Feedback

1: **Input:** $\{\eta^k\}_{k=0}^T > 0$, $x_0$, $e_i^0 = 0 \ \forall i \in [n]$
2: **for** $k = 0, 1, \ldots T$ **do**
3:    **Parallel: Worker side**
4:    **for** $i = 1, \ldots, n$ **do**
5:      obtain $g_i^k$
6:      send $\Delta_i^k = \mathcal{C}^k(\eta^k g_i^k + e_i^k)$ to master
7:      $e_i^{k+1} = \eta^k g_i^k + e_i^k - \Delta_i^k$
8:    **end for**
9:    **Master side**
10:   aggregate $\Delta^k = \frac{1}{n} \sum_{i=1}^n \Delta_i^k$
11:   broadcast $\Delta^k$ to each worker
12:   **Parallel: Worker side**
13:   **for** $i = 1, \ldots, n$ **do**
14:     $x^{k+1} = x^k - \Delta^k$
15:   **end for**
16: **end for**

• **Experimental Validation.** Finally, we provide an experimental evaluation on an array of classification tasks with CIFAR10 dataset corroborating our theoretical findings.

## 2    ERROR FEEDBACK IS NOT A GOOD IDEA WHEN USING UNBIASED COMPRESSORS

In this section we first introduce the notions of unbiased and general compression operators, and then compare Distributed Compressed SGD (DCSGD) without (Algorithm 1) and with (Algorithm 2) Error Feedback.

**Unbiased vs General Compression Operators.** We start with the definition of unbiased and general compression operators (Cordonnier, 2018; Stich et al., 2018; Koloskova et al., 2019).

**Definition 1** (Unbiased Compression Operator). A randomized mapping $\mathcal{C}: \mathbb{R}^d \to \mathbb{R}^d$ is an *unbiased compression operator (unbiased compressor)* if there exists $\delta \geq 1$ such that

$$\mathrm{E}\left[\mathcal{C}(x)\right] = x, \qquad \mathrm{E}\left\|\mathcal{C}(x)\right\|^2 \leq \delta \left\|x\right\|^2, \qquad \forall x \in \mathbb{R}^d. \tag{2}$$

If this holds, we will for simplicity write $\mathcal{C} \in \mathbb{U}(\delta)$.

**Definition 2** (General Compression Operator). A (possibly) randomized mapping $\mathcal{C}: \mathbb{R}^d \to \mathbb{R}^d$ is a *general compression operator (general compressor)* if there exists $\lambda > 0$ and $\delta \geq 1$ such that

$$\mathrm{E}\left[\left\|\lambda \mathcal{C}(x) - x\right\|^2\right] \leq \left(1 - \tfrac{1}{\delta}\right)\left\|x\right\|^2, \qquad \forall x \in \mathbb{R}^d. \tag{3}$$

If this holds, we will for simplicity write $\mathcal{C} \in \mathbb{C}(\delta)$.

The following lemma provides a link between these notions (see, e.g. Beznosikov et al. (2020)).

**Lemma 1.** *If $\mathcal{C} \in \mathbb{U}(\delta)$, then* (3) *holds with $\lambda = \frac{1}{\delta}$, i.e., $\mathcal{C} \in \mathbb{C}(\delta)$. That is, $\mathbb{U}(\delta) \subset \mathbb{C}(\delta)$.*

Note that the opposite inclusion to that established in the above lemma does not hold. For instance, the Top-$K$ operator belongs to $\mathbb{C}(\delta)$, but does not belong to $\mathbb{U}(\delta)$. In the next section we develop a procedure for transforming any mapping $\mathcal{C}: \mathbb{R}^d \to \mathbb{R}^d$ (and in particular, any general compressor) into a closely related *induced* unbiased compressor.

**Distributed SGD with vs without Error Feedback.** In the rest of this section, we compare the convergence rates for DCSGD (Algorithm 1) and DCSGD with EF (Algorithm 2). We do this comparison under standard assumptions (Karimi et al., 2016; Bottou et al., 2018; Necoara et al., 2019; Gower et al., 2019; Stich, 2019b; Stich & Karimireddy, 2020), listed next.

First, we assume throughout that $f$ has a unique minimizer $x^\star$, and let $f^\star = f(x^\star) > -\infty$.

**Assumption 1** ($\mu$-quasi convexity). $f$ is $\mu$-quasi convex, i.e.,

$$f^\star \geq f(x) + \langle \nabla f(x), x^\star - x \rangle + \frac{\mu}{2} \|x^\star - x\|^2, \qquad \forall x \in \mathbb{R}^d. \tag{4}$$

**Assumption 2** (unbiased gradient oracle). The stochastic gradient used in Algorithms 1 and 2 satisfies

$$\mathrm{E}\left[g_i^k \mid x^k\right] = \nabla f_i(x^k), \qquad \forall i, k. \tag{5}$$

Note that this assumption implies $\mathrm{E}\left[\frac{1}{n}\sum_{i=1}^n g_i^k \mid x^k\right] = \nabla f(x^k)$.

**Assumption 3** (($L, \sigma^2$)-expected smoothness). Function $f$ is $(L, \sigma^2)$-smooth if there exist constants $L > 0$ and $\sigma^2 \geq 0$ such that $\forall i \in [n]$ and $\forall x^k \in \mathbb{R}^d$

$$\mathrm{E}\left[\left\|g_i^k\right\|^2\right] \leq 2L(f_i(x^k) - f_i^\star) + \sigma^2, \tag{6}$$

$$\mathrm{E}\left[\left\|\frac{1}{n}\sum_{i=1}^n g_i^k\right\|^2\right] \leq 2L(f(x^k) - f^\star) + \sigma^2/n, \tag{7}$$

where $f_i^\star$ is the minimum functional value of $f_i$ and $[n] = \{1, 2, \dots, n\}$.

This assumption generalizes standard smoothness and boundedness of variance assumptions. For more details and discussion, see the works of Gower et al. (2019); Stich (2019b). Equipped with these assumptions, we are ready to proceed with the convergence theory.

**Theorem 2** (Convergence of DCSGD). *Consider the DCSGD algorithm with $n \geq 1$ nodes. Let Assumptions 1–3 hold and $\mathcal{C} \in \mathbb{U}(\delta)$, where $\delta_n = \frac{\delta-1}{n} + 1$. Let $D \coloneqq \frac{2L}{n}\sum_{i=1}^n (f_i(x^\star) - f_i^\star)$. Then there exist stepsizes $\eta^k \leq \frac{1}{2\delta_n L}$ and weights $w^k \geq 0$ such that for all $T \geq 1$ we have*

$$\mathrm{E}\left[f(\bar{x}^T) - f^\star\right] + \mu\mathrm{E}\left[\left\|x^T - x^\star\right\|^2\right] \leq 64\delta_n L r^0 \exp\left[-\frac{\mu T}{4\delta_n L}\right] + 36\frac{(\delta_n-1)D + \delta\sigma^2/n}{\mu T},$$

*where $r^0 = \left\|x^0 - x^\star\right\|^2$, $W^T = \sum_{k=0}^T w^k$, and $\mathrm{Prob}(\bar{x}^T = x^k) = w^k/W^T$.*

If $\delta = 1$ (no compression), Theorem 2 recovers the optimal rate of Distributed SGD (Stich, 2019b). If $\delta > 1$, there is an extra term $(\delta_n - 1)D$ in the convergence rate, which appears due to heterogenity of data ($\sum_{i=1}^n \nabla f_i(x^\star) = 0$, but $\sum_{i=1}^n \mathcal{C}(\nabla f_i(x^\star)) \neq 0$ in general). In addition, the rate is negatively affected by extra variance due to presence of compression which leads to $L \to \delta_n L$ and $\sigma^2/n \to \delta\sigma^2/n$.

Next we compare our rate to the best-known result for Error Feedback (Stich & Karimireddy, 2020) ($n = 1$), (Beznosikov et al., 2020) ($n \geq 1$) used with $\mathcal{C} \in \mathbb{U}(\delta) \subset \mathbb{C}(\delta)$

$$\mathrm{E}\left[f(\bar{x}^T) - f^\star\right] = \tilde{\mathcal{O}}\left(\delta L r^0 \exp\left[-\frac{\mu T}{\delta L}\right] + \frac{\delta D + \sigma^2}{\mu T}\right)$$

One can note several disadvantages of Error Feedback (Alg. 2) with respect to plain DCSGD (Alg. 1). The first major drawback is that the effect of compression $\delta$ is not reduced with an increasing number of nodes. Another disadvantage is that Theorem 2 implies convergence for both the functional values and the last iterate, rather than for functional values only as it is the case for EF. On top of that, our rate of DCSGD as captured by Theorem 2 does not contain any hidden polylogarithmic factor comparing to EF. Another practical supremacy of DCSGD is that there is no need to store an extra vector for the error, which reduces the storage costs by a factor of two, making Algorithm 1 a viable choice for Deep Learning models with millions of parameters. Finally, one does not need to assume standard $L$-smoothness in order to prove convergence in Theorem 2, while, one the other hand, $L$-smoothness is an important building block for proving convergence for general compressors due to the presence of bias (Stich & Karimireddy, 2020; Beznosikov et al., 2020). The only term in which EF might outperform plain DCSGD is $\mathcal{O}(\sigma^2/\mu T)$ for which the corresponding term is $\mathcal{O}(\delta\sigma^2/n\mu T)$. This is due to the fact that EF compensates for the error, while standard compression introduces extra variance. Note that this is not major issue as it is reasonable to assume $\delta/n = \mathcal{O}(1)$ or, in addition, $\sigma^2 = 0$ if weak growth condition holds (Vaswani et al., 2019), which is quite standard assumption, or one can remove effect of $\sigma^2$ by either computing full gradient locally or by incorporating variance reduction such as SVRG (Johnson & Zhang, 2013). In Section 4, we also discuss the way how to remove the effect of $D$ in Theorem 2. Putting all together, this suggests that standard DCSGD (Algorithm 1) is strongly preferable, in theory, to DCSGD with Error Feedback (Algorithm 2) for $\mathcal{C} \in \mathbb{U}(\delta)$.

## 3 INDUCED COMPRESSOR: FIXING BIAS WITH ERROR-COMPRESSION

In the previous section, we showed that compressed DCSGD is theoretically preferable to DCSGD with Error Feedback for $\mathcal{C} \in \mathbb{U}(\delta)$. Unfortunately, $\mathbb{C}(\delta) \not\subset \mathbb{U}(\delta)$, an example being the Top-$K$ compressor (Alistarh et al., 2018; Stich et al., 2018). This compressors belongs to $\mathbb{C}(\frac{d}{K})$, but does not belong to $\mathbb{U}(\delta)$ for any $\delta$. On the other hand, multiple unbiased alternatives to Top-$K$ have been proposed in the literature, including gradient sparsification (Wangni et al., 2018) and adaptive random sparsification (Beznosikov et al., 2020).

**Induced Compressor.** We now propose a *general mechanism for constructing an unbiased compressor $\mathcal{C} \in \mathbb{U}$ from any biased compressor $\mathcal{C}_1 \in \mathbb{C}$.* We shall argue that it is preferable to use this *induced compressor* within DCSGD, in both theory and practice, to using the original biased compressor $\mathcal{C}_1$ within DCSGD + Error Feedback.

**Theorem 3.** *For $\mathcal{C}_1 \in \mathbb{C}(\delta_1)$ with $\lambda = 1$, choose $\mathcal{C}_2 \in \mathbb{U}(\delta_2)$ and define the induced compressor via*

$$\mathcal{C}(x) \coloneqq \mathcal{C}_1(x) + \mathcal{C}_2(x - \mathcal{C}_1(x)).$$

*The induced compression operator satisfies $\mathcal{C} \in \mathbb{U}(\delta)$ with $\delta = \delta_2 \left(1 - 1/\delta_1\right) + 1/\delta_1$.*

To get some intuition about this procedure, recall the structure used in Error Feedback. The gradient estimator is first compressed with $\mathcal{C}_1(g)$ and the error $e = g - \mathcal{C}_1(g)$ is stored in memory and used to modify the gradient in the next iteration. In our proposed approach, instead of storing the error $e$, we compress it with an unbiased compressor $\mathcal{C}_2$ (which can be seen as a parameter allowing flexibility in the design of the induced compressor) and communicate *both* of these compressed vectors. Note that this procedure results in extra variance as we do not work with the exact error, but with its unbiased estimate only. On the other hand, there is no bias and error accumulation that one needs to correct for. In addition, due to our construction, at least the same amount of information is sent to the master as in the case of plain $\mathcal{C}_1(g)$: indeed, we send both $\mathcal{C}_1(g)$ and $\mathcal{C}_2(e)$. The drawback of this is the necessity to send more bits. However, Theorem 3 provides the freedom in generating the induced compressor through the choice of the unbiased compressor $\mathcal{C}_2$. In theory, it makes sense to choose $\mathcal{C}_2$ with similar compression factor to the compressor $\mathcal{C}_1$ we are transforming as this way the total number of communicated bits per iteration is preserved, up to the factor of two.

*Remark:* The $\text{rtop}_{k_1,k_2}(x, y)$ operator proposed by Elibol et al. (2020) can be seen as a special case of our induced compressor with $x = y$, $\mathcal{C}_1 = \text{Top-}k_1$ and $\mathcal{C}_2 = \text{Rand-}k_2$.

**Benefits of Induced Compressor.** In the light of the results in Section 2, we argue that one should always prefer unbiased compressors to biased ones as long as their variances $\delta$ and communication complexities are the same, e.g., Rand-$K$ over Top-$K$. In practice, biased/greedy compressors are in some settings observed to perform better due to their lower empirical variance (Beznosikov et al., 2020). These considerations give a practical significance to Theorem 3 as we demonstrate on the following example. Let us consider two compressors: one biased $\mathcal{C}_1 \in \mathbb{C}(\delta_1)$ and one unbiased $\mathcal{C}_2 \in \mathbb{U}(\delta_2)$, such that $\delta_1 = \delta_2 = \delta$, having identical communication complexity, e.g., Top-$K$ and Rand-$K$. The induced compressor $\mathcal{C}(x) \coloneqq \mathcal{C}_1(x) + \mathcal{C}_2(x - \mathcal{C}_1(x))$ belongs to $\mathbb{U}(\delta_3)$, where $\delta_3 = \delta - \left(1 - \frac{1}{\delta}\right) < \delta$. While the size of the transmitted message is doubled, one can use Algorithm 1 since $\mathcal{C}$ is unbiased, which provides better convergence guarantees than Algorithm 2. Based on the construction of the induced compressor, one might expect that we need extra memory as "the error" $e = g - \mathcal{C}_1(g)$ needs to be stored, but during computation only. This is not an issue as compressors for DNNs are always applied layer-wise (Dutta et al., 2019), and hence the size of the extra memory is negligible. It does not help EF, as the error needs to be stored at any time for each layer.

## 4 EXTENSIONS

We now develop several extensions of Algorithm 1 relevant to distributed optimization in general, and to Federated Learning in particular. This is all possible due to the simplicity of our approach. Note that in the case of Error Feedback, these extensions have either not been obtained yet, or similarly to Section 2, the results are worse when compared to our derived bounds for unbiased compressors.

**Partial Participation with Arbitrary Distribution over Nodes.** In this section, we extend our results to a variant of DCSGD utilizing *partial participation*, which is of key relevance to Federated

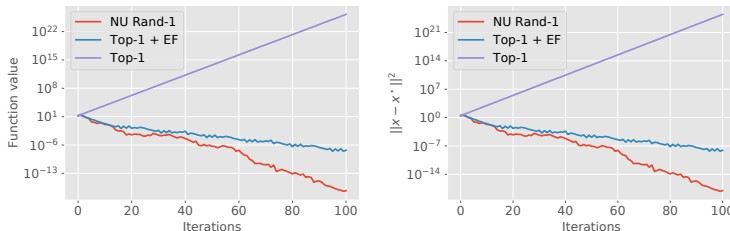

**Figure 1:** Comparison of Top-1 (+ EF) and NU Rand-1 on Example 1 from Beznosikov et al. (2020).

Learning. In this framework, only a subset of all nodes communicates to the master node in each communication round. Such framework was analyzed before, but only for the case of uniform subsampling (Sattler et al., 2019; Reisizadeh et al., 2020). In our work, we consider a more general partial participation framework: we assume that the subset of participating clients is determined by a fixed but otherwise arbitrary random set-valued mapping $\mathbb{S}$ (a "sampling") with values in $2^{[n]}$, where $[n] = \{1, 2, \ldots, n\}$. To the best of our knowledge, this is the first partial participation result for FL where an arbitrary distribution over the nodes is considered. On the other hand, this is not the first work which makes use of the arbitrary sampling paradigm; this was used before in other contexts, e.g., for obtaining importance sampling guarantees for coordinate descent (Qu et al., 2015), primal-dual methods (Chambolle et al., 2018), and variance reduction (Horváth & Richtárik, 2019).

Note that the sampling $\mathbb{S}$ is uniquely defined by assigning probabilities to all $2^n$ subsets of $[n]$. With each sampling $\mathbb{S}$ we associate a *probability matrix* $\mathbf{P} \in \mathbb{R}^{n \times n}$ defined by $\mathbf{P}_{ij} := \text{Prob}(\{i, j\} \subseteq \mathbb{S})$. The *probability vector* associated with $\mathbb{S}$ is the vector composed of the diagonal entries of $\mathbf{P}$: $p = (p_1, \ldots, p_n) \in \mathbb{R}^n$, where $p_i := \text{Prob}(i \in \mathbb{S})$. We say that $\mathbb{S}$ is *proper* if $p_i > 0$ for all $i$. It is easy to show that $b := \text{E}[|\mathbb{S}|] = \text{Trace}(\mathbf{P}) = \sum_{i=1}^{n} p_i$, and hence $b$ can be seen as the expected number of clients participating in each communication round.

There are two algorithmic changes due to this extension: line 4 of Algorithm 1 does not iterate over every node, only over nodes $i \in S^k$, where $S^k \sim \mathbb{S}$, and the aggregation step in line 9 is adjusted to lead to an unbiased estimator of the gradient, which gives $\Delta_k = \sum_{i \in S^k} \frac{1}{np_i} \Delta_i^k$.

To prove convergence, we exploit the following lemma.

**Lemma 4** (Lemma 1, Horváth & Richtárik (2019)). *Let $\zeta_1, \zeta_2, \ldots, \zeta_n$ be vectors in $\mathbb{R}^d$ and let $\bar{\zeta} := \frac{1}{n} \sum_{i=1}^{n} \zeta_i$ be their average. Let $\mathbb{S}$ be a proper sampling. Then there exists $v \in \mathbb{R}^n$ such*

$$\mathbf{P} - pp^{\top} \preceq \mathbf{Diag}(p_1 v_1, p_2 v_2, \ldots, p_n v_n). \tag{8}$$

*Moreover, if $S \sim \mathbb{S}$, then*

$$\text{E}\left[\left\|\sum_{i \in S} \frac{\zeta_i}{np_i} - \bar{\zeta}\right\|^2\right] \leq \frac{1}{n^2} \sum_{i=1}^{n} \frac{v_i}{p_i} \|\zeta_i\|^2. \tag{9}$$

The following theorem establishes the convergence rate for Algorithm 1 with partial participation.

**Theorem 5.** *Let Assumptions 1–3 hold and $\mathcal{C} \in \mathbb{U}(\delta)$, then there exist stepsizes $\eta^k \leq \frac{1}{2\delta_{\mathbb{S}} L}$ and weights $w^k \geq 0$ such that*

$$\text{E}\left[f(\bar{x}^T) - f^{\star}\right] + \mu \text{E}\left[\left\|x^T - x^{\star}\right\|^2\right] \leq 64\delta_{\mathbb{S}} L r^0 \exp\left[-\frac{\mu T}{4\delta_{\mathbb{S}} L}\right] + 36\frac{(\delta_{\mathbb{S}} - 1)D + (1 + a_{\mathbb{S}})\delta\sigma^2/n}{\mu T},$$

*where $r^0, W^T, \bar{x}^T$, and $D$ are defined in Theorem 2, $a_{\mathbb{S}} = \max_{i \in [n]}\{v_i/p_i\}$, and $\delta_{\mathbb{S}} = \frac{\delta a_{\mathbb{S}} + (\delta - 1)}{n} + 1$.*

For the case $\mathbb{S} = [n]$ with probability 1, one can show that Lemma 4 holds with $v = 0$, and hence we exactly recover the results of Theorem 2. In addition, we can quantify the slowdown factor with respect to full participation regime (Theorem 2), which is $\delta \max_{i \in [n]} \frac{v_i}{p_i}$. While in our framework we assume the distribution $\mathbb{S}$ to be fixed, it can be easily extended to several proper distributions $\mathbb{S}_j$'s or we can even handle a block-cyclic structure with each block having an arbitrary proper distribution $\mathbb{S}_j$ over the given block $j$ combining our analysis with the results of Eichner et al. (2019).

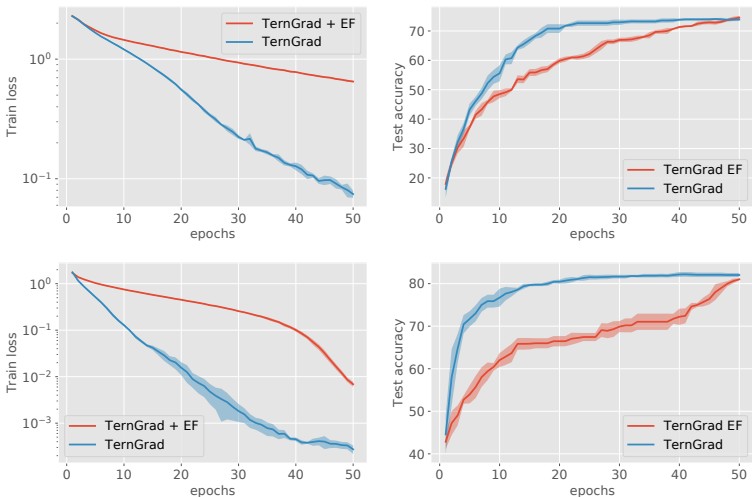

**Figure 2:** Algorithm 1 vs. Algorithm 2 on CIFAR10 with ResNet18 (bottom), VGG11 (top) and TernGrad as a compression.

**Obtaining Linear Convergence.** Note that in all the previous theorems, we can only guarantee a sublinear $\mathcal{O}(1/T)$ convergence rate. Linear rate is obtained in the special case when $D = 0$ and $\sigma^2 = 0$. The first condition is satisfied, when $f_i^\star = f_i(x^\star)$ for all $i \in [n]$, thus when $x^\star$ is also minimizer of every local function $f_i$. Furthermore, the effect od $D$ can be removed using compression of gradient differences, as pioneered in the DIANA algorithm (Mishchenko et al., 2019a). Note that $\sigma^2 = 0$ if weak growth condition holds (Vaswani et al., 2019). Moreover, one can remove effect of $\sigma^2$ by either computing full gradients locally or by incorporating variance reduction such as SVRG (Johnson & Zhang, 2013). It was shown by Horváth et al. (2019b) that both $\sigma^2$ and $D$ can be removed for the setting of Theorem 2. These results can be easily extended to partial participation using our proof technique for Theorem 5. Note that this reduction is not possible for Error Feedback as the analysis of the DIANA algorithm is heavily dependent on the unbiasedness property. This points to another advantage of the induced compressor framework introduced in Section 3.

**Acceleration.** We now comment on the combination of compression and acceleration/momentum. This setting is very important to consider as essentially all state-of-the-art methods for training deep learning models, including Adam (Kingma & Ba, 2015; Reddi et al., 2018), rely on the use of momentum in one form or another. One can treat the unbiased compressed gradient as a stochastic gradient (Gorbunov et al., 2020) and the theory for momentum SGD (Yang et al., 2016; Gadat et al., 2018; Loizou & Richtárik, 2017) would be applicable with an extra smoothness assumption. Moreover, it is possible to remove the variance caused by stochasticity and obtain linear convergence with an accelerated rate, which leads to the Accelerated DIANA method (Li et al., 2020). Similarly to our previous discussion, both of these techniques are heavily dependent on the unbiasedness property. It is an intriguing question, but out of the scope of the paper, to investigate the combined effect of momentum and Error Feedback and see whether these techniques are compatible theoretically.

## 5 EXPERIMENTS

In this section, we compare Algorithms 1 and 2 for several compression operators. If the method contains " + EF ", it means that EF is applied, thus Algorithm 2 is applied. Otherwise, Algorithm 1 is displayed. To be fair, we always compare methods with the same communication complexity per iteration. All experimental details can be found in the Appendix.

**Failure of DCSGD with biased Top-1.** In this experiment, we present example considered in Beznosikov et al. (2020), which was used as a counterexample to show that some form of error correction is needed in order for biased compressors to work/provably converge. In addition, we run experiments on their construction and show that while Error Feedback fixes divergence, it is still significantly dominated by unbiased non-uniform sparsification(NU Rand-1), which works by only

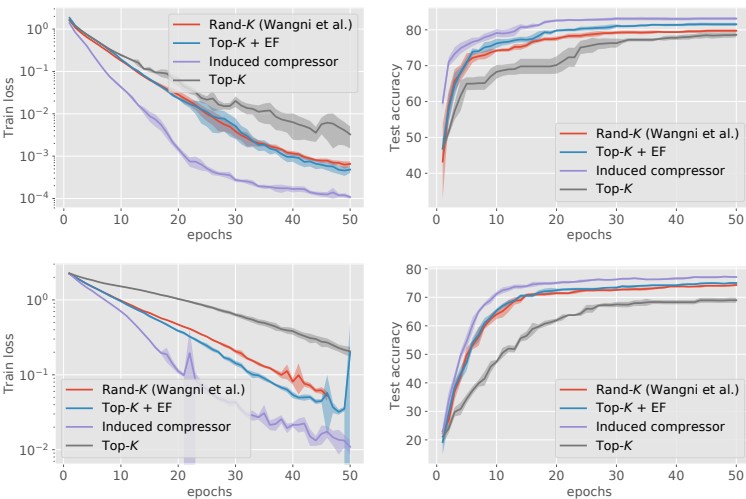

**Figure 3:** Comparison of different sparsification techniques with and without usage of Error Feedback on CIFAR10 with Resnet18 (top) and VGG11 (bottom). $K = 5\% * d$, for Induced compressor $\mathcal{C}_1$ is Top-$K/2$ and $\mathcal{C}_2$ is Rand-$K/2$ (Wangni et al.).

keeping one non-zero coordinate sampled with probability equal to $|x|/\sum_{i=1}^{d} |x|_i$, where $|x|$ denotes element-wise absolute value, as can be seen in Figure 1. The details can be found in the Appendix.

**Error Feedback for Unbiased Compression Operators.** In our second experiment, we compare the effect of Error Feedback in the case when an unbiased compressor is used. Note that unbiased compressors are theoretically guaranteed to work both with Algorithm 1 and 2. We can see from Figure 2 that adding Error Feedback can hurt the performance; we use TernGrad (Wen et al., 2017) (coincides with QSGD (Alistarh et al., 2016) and natural dithering (Horváth et al., 2019a) with the infinity norm and one level) as compressors. This agrees with our theoretical findings. In addition, for sparsification techniques such as Random Sparsification or Gradient Sparsification (Wangni et al., 2018), we observed that when sparsity is set to be 10 %, Algorithm 1 converges for all the selected values of step-sizes, but Algorithm 2 diverges and a smaller step-size needs to be used. This is an important observation as many practical works (Li et al., 2014; Wei et al., 2015; Aji & Heafield, 2017; Hsieh et al., 2017; Lin et al., 2018b; Lim et al., 2018) use sparsification techniques mentioned in this section, but proposed to use EF, while our work shows that using unbiasedness property leads not only to better convergence but also to memory savings.

**Unbiased Alternatives to Biased Compression.** In this section, we investigate candidates for unbiased compressors than can compete with Top-$K$, one of the most frequently used compressors. Theoretically, Top-$K$ is not guaranteed to work by itself and might lead to divergence (Beznosikov et al., 2020) unless Error Feedback is applied. One would usually compare the performance of Top-$K$ with EF to Rand-$K$, which keeps $K$ randomly selected coordinates and then scales the output by $d/K$ to preserve unbiasedness. Rather than naively comparing to Rand-$K$, we propose to use more nuanced unbiased approaches. The first one is Gradient Sparsification proposed by Wagni et al. (Wangni et al., 2018), which we refer to here as Rand-$K$ (Wangni et al.), where the probability of keeping each coordinate scales with its magnitude and communication budget. As the second alternative, we propose to use our induced compressor, where $\mathcal{C}_1$ is Top-$a$ and unbiased part $\mathcal{C}_2$ is Rand-$(K - a)$ (Wangni et al.) with communication budget $K - a$. It should be noted that $a$ can be considered as a hyperparameter to tune. For our experiment, we chose it to be $K/2$ for simplicity. Figure 3 suggests that our induced compressor outperforms all of its competitors as can be seen for both VGG11 and Resnet18. Moreover, induced compressor as well as Rand-$K$ do not require extra memory to store the error vector. Finally, Top-$K$ without EF suffers a significant decrease in performance, which stresses the necessity of error correction.

## 6 CONCLUSION

In this paper, we argue that if compressed communication is required for distributed training due to communication overhead, it is better to use unbiased compressors. We show that this leads to strictly better convergence guarantees with fewer assumptions. In addition, we propose a new construction for transforming any compressor into an unbiased one using a compressed EF-like approach. Besides theoretical superiority, usage of unbiased compressors enjoys lower memory requirements. Our theoretical findings are corroborated with empirical evaluation.

As a future work we plan to investigate the question of the appropriate choice of the inducing compressor $\mathcal{C}$. Our preliminary studies show that there is much to be discovered here, both in theory and in terms of developing further practical guidelines to those already contained in this work. The question of (theoretically) optimizing for $\mathcal{C}_1$ and $\mathcal{C}_2$ is difficult, as it necessitates a deeper theoretical understanding of biased compressors, which is currently missing. An alternative is to impose some assumptions on the structure of gradients encountered during the iterative process, or to perform an extensive experimental evaluation on desired tasks to provide guidelines for practitioners.

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

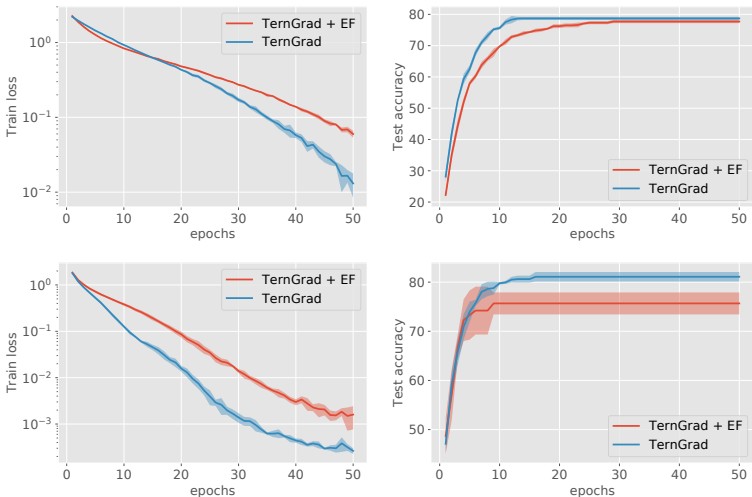

**Figure 4:** Algorithm 1 vs. Algorithm 2 on CIFAR10 with ResNet18 (bottom), VGG11 (top) and TernGrad as a compression.

## APPENDIX

## A    EXPERIMENTAL DETAILS

To be fair, we always compare methods with the same communication complexity per iteration. We report the number of epochs (passes over the dataset) with respect to training loss and testing accuracy. The test accuracy is obtained by evaluating the best model in terms of validation accuracy. A validation accuracy is computed based on 10 % randomly selected training data. We tune the step-size using based on the training loss. For every experiment, we randomly distributed the training dataset among 8 workers; each worker computes its local gradient-based on its own dataset. We used a local batch size of 32. All the provided figures display the mean performance with one standard error over 5 independent runs. For a fair comparison, we use the same random seed for the compared methods. Our experimental results are based on a Python implementation of all the methods running in PyTorch. All reported quantities are independent of the system architecture and network bandwidth.

**Dataset and Models.** We do an evaluation on CIFAR10 dataset. We consider VGG11 (Simonyan & Zisserman, 2015) and ResNet18 (He et al., 2016) models and step-sizes $0.1, 0.05$ and $0.01$.

### A.1    EXTRA EXPERIMENTS

**Momentum.** In this extra experiment, we look at the effect of momentum on Algorithm 1 and 2. We set momentum to $0.9$. Similarly to Figure 2, we work with the unbiased compressor, concretely Tern-Grad (Wen et al., 2017) (coincides with QSGD (Alistarh et al., 2016) and natural dithering (Horváth et al., 2019a) with the infinity norm and one level), to see the effect of adding Error Feedback. We can see from Figure 4 that adding Error Feedback can hurt the performance, which agrees with our theoretical findings.

## B    EXAMPLE 1, BEZNOSIKOV ET AL. (2020)

In this section, we present example considered in Beznosikov et al. (2020), which was used as a counterexample to show that some form of error correction is needed in order for biased compressors to work/provably converge. In addition, we run experiments on their construction and show that while Error Feedback fixes divergence, it is still significantly dominated by unbiased non-uniform sparsification as can be seen in Figure 1. The construction follows.

Consider $n = d = 3$ and define the following smooth and strongly convex quadratic functions

$$f_1(x) = \langle a, x \rangle^2 + \frac{1}{4} \|x\|^2, \qquad f_2(x) = \langle b, x \rangle^2 + \frac{1}{4} \|x\|^2, \qquad f_3(x) = \langle c, x \rangle^2 + \frac{1}{4} \|x\|^2,$$

where $a = (-3, 2, 2), b = (2, -3, 2), c = (2, 2, -3)$. Then, with the initial point $x^0 = (t, t, t), \ t > 0$

$$\nabla f_1(x^0) = \frac{t}{2}(-11, 9, 9), \qquad \nabla f_2(x^0) = \frac{t}{2}(9, -11, 9), \qquad \nabla f_3(x^0) = \frac{t}{2}(9, 9, -11).$$

Using the Top-1 compressor, we get

$$\mathcal{C}(\nabla f_1(x^0)) = \tfrac{t}{2}(-11, 0, 0), \quad \mathcal{C}(\nabla f_2(x^0)) = \tfrac{t}{2}(0, -11, 0), \quad \mathcal{C}(\nabla f_3(x^0)) = \tfrac{t}{2}(0, 0, -11).$$

The next iterate of DCGD is

$$x^1 = x^0 - \frac{\eta}{3} \sum_{i=1}^{3} \mathcal{C}(\nabla f_i(x^0)) = \left(1 + \frac{11\eta}{6}\right) x^0.$$

Repeated application gives $x^k = \left(1 + \frac{11\eta}{6}\right)^k x^0$, which diverges exponentially fast to $+\infty$ since $\eta > 0$.

As a initial point, we use $(1, 1, 1)^\top$ in our experiments and we choose step size $\frac{1}{L}$, where $L$ is smoothness parameter of $f = \frac{1}{3}(f_1 + f_2 + f_3)$. Note that zero vector is the unique minimizer of $f$.

## C  PROOFS

### C.1  PROOF OF LEMMA 1

We follow (2), which holds for $\mathcal{C} \in \mathbb{U}(\delta)$.

$$\begin{aligned}
\mathrm{E}\left[\left\|\frac{1}{\delta}\mathcal{C}^k(x) - x\right\|^2\right] &= \frac{1}{\delta^2}\mathrm{E}\left[\|\mathcal{C}^k(x)\|^2\right] - 2\frac{1}{\delta}\left\langle \mathrm{E}\left[\mathcal{C}^k(x)\right], x\right\rangle + \|x\|^2 \\
&\leq \left(\frac{1}{\delta} - \frac{2}{\delta} + 1\right)\|x\|^2 \\
&= \left(1 - \frac{1}{\delta}\right)\|x\|^2,
\end{aligned}$$

which concludes the proof.

## C.2 PROOF OF THEOREM 2

We use the update of Algorithm 1 to bound the following quantity

$$
\begin{aligned}
\mathrm{E}\left[\left\|x^{k+1}-x^\star\right\|^2 |x^k\right] &= \left\|x^k-x^\star\right\|^2 - \frac{\eta^k}{n}\sum_{i=1}^{n}\mathrm{E}\left[\langle \mathcal{C}^k(g_i^k), x^k-x^\star\rangle |x^k\right] + \\
&\quad \left(\frac{\eta^k}{n}\right)^2 \mathrm{E}\left[\left\|\sum_{i=1}^{n}\mathcal{C}^k(g_i^k)\right\|^2 |x^k\right] \\
&\overset{(2)+(5)}{\leq} \left\|x^k-x^\star\right\|^2 - \eta^k\langle\nabla f(x^k), x^k-x^\star\rangle + \\
&\quad \frac{(\eta^k)^2}{n^2}\mathrm{E}\left[\sum_{i=1}^{n}\left\|\mathcal{C}^k(g_i^k)-g_i^k\right\|^2 + \left\|\sum_{i=1}^{n}g_i^k\right\|^2 |x^k\right] \\
&\overset{(2)}{\leq} \left\|x^k-x^\star\right\|^2 - \eta^k\langle\nabla f(x^k), x^k-x^\star\rangle + \\
&\quad \frac{(\eta^k)^2}{n^2}\mathrm{E}\left[(\delta-1)\sum_{i=1}^{n}\left\|g_i^k\right\|^2 + \left\|\sum_{i=1}^{n}g_i^k\right\|^2 |x^k\right] \\
&\overset{(6)+(7)}{\leq} \left\|x^k-x^\star\right\|^2 - \eta^k\langle\nabla f(x^k), x^k-x^\star\rangle + \\
&\quad 2L(\eta^k)^2\left(\delta_n(f(x^k)-f^\star)+(\delta_n-1)\frac{1}{n}\sum_{i=1}^{n}(f_i(x^\star)-f_i^\star)\right) + (\eta^k)^2\frac{\delta\sigma^2}{n} \\
&\overset{(4)}{\leq} (1-\mu\eta^k)\left\|x^k-x^\star\right\|^2 - 2\eta^k\left(1-\eta^k\delta_n L\right)(f(x^k)-f^\star) + \\
&\quad (\eta^k)^2\left((\delta_n-1)D+\frac{\delta\sigma^2}{n}\right).
\end{aligned}
$$

Taking full expectation and $\eta^k \leq \frac{1}{2\delta_n L}$, we obtain

$$
\mathrm{E}\left[\left\|x^{k+1}-x^\star\right\|^2\right] \leq (1-\mu\eta^k)\mathrm{E}\left[\left\|x^k-x^\star\right\|^2\right] - \eta^k\mathrm{E}\left[f(x^k)-f^\star\right] + (\eta^k)^2\left((\delta_n-1)D+\frac{\delta\sigma^2}{n}\right).
$$

The rest of the analysis is closely related to the one of Stich (2019b). We would like to point out that similar results to Stich (2019b) were also present in (Lacoste-Julien et al., 2012; Stich et al., 2018; Grimmer, 2019).

We first rewrite the previous inequality to the form

$$
r^{k+1} \leq (1-a\eta^k)r^k - \eta^k s^k + (\eta^k)^2 c, \tag{10}
$$

where $r^k = \mathrm{E}\left[\left\|x^k-x^\star\right\|^2\right]$, $s^k = \mathrm{E}\left[f(x^k)-f^\star\right]$, $a = \mu$, $c = (\delta_n-1)D+\frac{\delta\sigma^2}{n}$.

We proceed with lemmas that establish a convergence guarantee for every recursion of type (10).

**Lemma 6.** *Let $\{r^k\}_{k\geq 0}$, $\{s^k\}_{k\geq 0}$ be as in (10) for $a > 0$ and for constant stepsizes $\eta^k \equiv \eta := \frac{1}{d}$, $\forall k \geq 0$. Then it holds for all $T \geq 0$:*

$$
r^T \leq r^0\exp\left[-\frac{aT}{d}\right] + \frac{c}{ad}.
$$

*Proof.* This follows by relaxing (10) using $\mathrm{E}\left[f(x^k)-f^\star\right] \geq 0$, and unrolling the recursion

$$
r^T \leq (1-a\eta)r^{T-1}+c\gamma^2 \leq (1-a\eta)^T r^0 + c\eta^2\sum_{k=0}^{T-1}(1-a\eta)^k \leq (1-a\eta)^T r^0 + \frac{c\eta}{a}. \tag{11}
$$

$\square$

**Lemma 7.** *Let $\{r^k\}_{k \geq 0}$, $\{s^k\}_{k \geq 0}$ as in (10) for $a > 0$ and for decreasing stepsizes $\eta^k := \frac{2}{a(\kappa+k)}$, $\forall k \geq 0$, with parameter $\kappa := \frac{2d}{a}$, and weights $w^k := (\kappa + k)$. Then*

$$\frac{1}{W^T} \sum_{k=0}^{T} s^k w^k + a r^{T+1} \leq \frac{2a\kappa^2 r_0}{T^2} + \frac{2c}{aT},$$

*where $W^T := \sum_{k=0}^{T} w^k$.*

*Proof.* We start by re-arranging (10) and multiplying both sides with $w^k$

$$
\begin{aligned}
s^k w^k &\leq \frac{w^k(1 - a\eta^k)r^k}{\eta^k} - \frac{w^k r^{k+1}}{\eta^k} + c\eta^k w^k \\
&= a(\kappa + k)(\kappa + k - 2)r^k - a(\kappa + k)^2 r^{k+1} + \frac{c}{a} \\
&\leq a(\kappa + k - 1)^2 r^k - a(\kappa + k)^2 r^{k+1} + \frac{c}{a},
\end{aligned}
$$

where the equality follows from the definition of $\eta^k$ and $w^k$ and the inequality from $(\kappa + k)(\kappa + k - 2) = (\kappa + k - 1)^2 - 1 \leq (\kappa + k - 1)^2$. Again we have a telescoping sum:

$$\frac{1}{W^T} \sum_{k=0}^{T} s^k w^k + \frac{a(\kappa + T)^2 r^{T+1}}{W^T} \leq \frac{a\kappa^2 r_0}{W^T} + \frac{c(T+1)}{aW^T},$$

with

- $W^T = \sum_{k=0}^{T} w^k = \sum_{k=0}^{T}(\kappa + k) = \frac{(2\kappa + T)(T+1)}{2} \geq \frac{T(T+1)}{2} \geq \frac{T^2}{2}$,

- and $W^T = \frac{(2\kappa + T)(T+1)}{2} \leq \frac{2(\kappa + T)(1+T)}{2} \leq (\kappa + T)^2$ for $\kappa = \frac{2d}{a} \geq 1$.

By applying these two estimates we conclude the proof. $\qquad\square$

The convergence can be obtained as the combination of these two lemmas.

**Lemma 8.** *Let $\{r^k\}_{k \geq 0}$, $\{s^k\}_{k \geq 0}$ as in (10), $a > 0$. Then there exists stepsizes $\eta^k \leq \frac{1}{d}$ and weighs $w^k \geq 0$, $W^T := \sum_{k=0}^{T} w^k$, such that*

$$\frac{1}{W^T} \sum_{k=0}^{T} s^k w^k + a r^{T+1} \leq 32 d r_0 \exp\left[-\frac{aT}{2d}\right] + \frac{36c}{aT}.$$

*Proof of Lemma 8.* For integer $T \geq 0$, we choose stepsizes and weights as follows

$$
\begin{aligned}
&\text{if } T \leq \frac{d}{a}, &\eta^k &= \frac{1}{d}, &w^k &= (1 - a\eta^k)^{-(k+1)} = \left(1 - \frac{a}{d}\right)^{-(k+1)}, \\
&\text{if } T > \frac{d}{a} \text{ and } k < t_0, &\eta^k &= \frac{1}{d}, &w^k &= 0, \\
&\text{if } T > \frac{d}{a} \text{ and } k \geq t_0, &\eta^k &= \frac{2}{a(\kappa + k - t_0)}, &w^k &= (\kappa + k - t_0)^2,
\end{aligned}
$$

for $\kappa = \frac{2d}{a}$ and $t_0 = \lceil \frac{T}{2} \rceil$. We will now show that these choices imply the claimed result.

We start with the case $T \leq \frac{d}{a}$. For this case, the choice $\eta = \frac{1}{d}$ gives

$$
\begin{aligned}
\frac{1}{W^T} \sum_{k=0}^{T} s^k w^k + a r^{T+1} &\leq (1 - a\eta)^{(T+1)} \frac{r_0}{\eta} + c\eta \\
&\leq \frac{r_0}{\eta} \exp\left[-a\eta(T+1)\right] + c\eta \\
&\leq d r_0 \exp\left[-\frac{aT}{d}\right] + \frac{c}{aT}.
\end{aligned}
$$

If $T > \frac{d}{a}$, then we obtain from Lemma 6 that

$$r^{t_0} \le r^0 \exp\left[-\frac{aT}{2d}\right] + \frac{c}{ad} .$$

From Lemma 7 we have for the second half of the iterates:

$$\frac{1}{W^T} \sum_{k=0}^{T} s^k w^k + ar^{T+1} = \frac{1}{W^T} \sum_{k=t_0}^{T} s^k w^k + ar^{T+1} \le \frac{8a\kappa^2 r^{t_0}}{T^2} + \frac{4c}{aT} .$$

Now we observe that the restart condition $r^{t_0}$ satisfies:

$$\frac{a\kappa^2 r^{t_0}}{T^2} = \frac{a\kappa^2 r^0 \exp\left(-\frac{aT}{2d}\right)}{T^2} + \frac{\kappa^2 c}{dT^2} \le 4ar^0 \exp\left[-\frac{aT}{2d}\right] + \frac{4c}{aT} ,$$

because $T > \frac{d}{a}$. These conclude the proof.

$\square$

Having these general convergence lemmas for the recursion of the form (10), the proof of the theorem follows directly from Lemmas 6 and 8 with $a = \mu$, $c = \sigma^2$, $d = 2\delta_n L$ . It is easy to check that condition $\eta^k \le \frac{1}{d} = \frac{1}{2\delta_n L}$ is satisfied.

## C.3 PROOF OF THEOREM 3

We have to show that our new compression is unbiased and has bounded variance. We start with the first property with $\lambda = 1$.

$$\begin{aligned}
\mathrm{E}\left[\mathcal{C}_1(x) + \mathcal{C}_2(x - \mathcal{C}_1(x))\right] &= \mathrm{E}_{\mathcal{C}_1}\left[\mathrm{E}_{\mathcal{C}_2}\left[\mathcal{C}_1(x) + \mathcal{C}_2(x - \mathcal{C}_1(x))|\mathcal{C}_1(x)\right]\right] \\
&= \mathrm{E}_{\mathcal{C}_1}\left[\mathcal{C}_1(x) + x - \mathcal{C}_1(x)\right] = x,
\end{aligned}$$

where the first equality follows from tower property and the second from unbiasedness of $\mathcal{C}_2$. For the second property, we also use tower property

$$\begin{aligned}
\mathrm{E}\left[\|\mathcal{C}_1(x) - x + \mathcal{C}_2(x - \mathcal{C}_1(x))\|^2\right] &= \mathrm{E}_{\mathcal{C}_1}\left[\mathrm{E}_{\mathcal{C}_2}\left[\|\mathcal{C}_1(x) - x + \mathcal{C}_2(x - \mathcal{C}_1(x))\|^2 |\mathcal{C}_1(x)\right]\right] \\
&\le (\delta_2 - 1)\mathrm{E}_{\mathcal{C}_1}\left[\|\mathcal{C}_1(x) - x\|^2\right] \\
&\le (\delta_2 - 1)\left(1 - \frac{1}{\delta_1}\right)\|x\|^2 ,
\end{aligned}$$

where the first and second inequalities follow directly from (2) and (3).

## C.4 PROOF OF LEMMA 4 (HORVÁTH & RICHTÁRIK, 2019)

For the first part of the claim, it was shown that $\mathbf{P} - pp^\top$ is positive semidefinite (Richtárik & Takáč, 2016), thus we can bound $\mathbf{P} - pp^\top \preceq n\mathbf{Diag}\left(\mathbf{P} - pp^\top\right) = \mathbf{Diag}\left(p \circ v\right)$, where $v_i = n(1 - p_i)$, which implies that (8) holds for this choice of $v$.

For the second part of the claim, let $1_{i\in\mathbb{S}} = 1$ if $i \in \mathbb{S}$ and $1_{i\in\mathbb{S}} = 0$ otherwise. Likewise, let $1_{i,j\in\mathbb{S}} = 1$ if $i,j \in \mathbb{S}$ and $1_{i,j\in\mathbb{S}} = 0$ otherwise. Note that $\mathrm{E}\left[1_{i\in\mathbb{S}}\right] = p_i$ and $\mathrm{E}\left[1_{i,j\in\mathbb{S}}\right] = p_{ij}$. Next, let us compute the mean of $X := \sum_{i\in\mathbb{S}} \frac{\zeta_i}{np_i}$:

$$\mathrm{E}\left[X\right] = \mathrm{E}\left[\sum_{i\in\mathbb{S}} \frac{\zeta_i}{np_i}\right] = \mathrm{E}\left[\sum_{i=1}^{n} \frac{\zeta_i}{np_i} 1_{i\in\mathbb{S}}\right] = \sum_{i=1}^{n} \frac{\zeta_i}{np_i}\mathrm{E}\left[1_{i\in\mathbb{S}}\right] = \frac{1}{n}\sum_{i=1}^{n} \zeta_i = \bar{\zeta}. \qquad (12)$$

Let $\mathbf{A} = [a_1, \ldots, a_n] \in \mathbb{R}^{d \times n}$, where $a_i = \frac{\zeta_i}{p_i}$, and let $e$ be the vector of all ones in $\mathbb{R}^n$. We now write the variance of $X$ in a form which will be convenient to establish a bound:

$$
\begin{aligned}
\mathrm{E}\left[\|X - \mathrm{E}[X]\|^2\right] &= \mathrm{E}\left[\|X\|^2\right] - \|\mathrm{E}[X]\|^2 \\
&= \mathrm{E}\left[\left\|\sum_{i \in \mathbb{S}} \frac{\zeta_i}{np_i}\right\|^2\right] - \|\bar{\zeta}\|^2 \\
&= \mathrm{E}\left[\sum_{i,j} \frac{\zeta_i^\top}{np_i} \frac{\zeta_j}{np_j} 1_{i,j \in \mathbb{S}}\right] - \|\bar{\zeta}\|^2 \\
&= \sum_{i,j} p_{ij} \frac{\zeta_i^\top}{np_i} \frac{\zeta_j}{np_j} - \sum_{i,j} \frac{\zeta_i^\top}{n} \frac{\zeta_j}{n} \\
&= \frac{1}{n^2} \sum_{i,j} (p_{ij} - p_i p_j) a_i^\top a_j \\
&= \frac{1}{n^2} e^\top \left((\mathbf{P} - pp^\top) \circ \mathbf{A}^\top \mathbf{A}\right) e. \qquad (13)
\end{aligned}
$$

Since by assumption we have $\mathbf{P} - pp^\top \preceq \mathbf{Diag}(p \circ v)$, we can further bound

$$
e^\top \left((\mathbf{P} - pp^\top) \circ \mathbf{A}^\top \mathbf{A}\right) e \leq e^\top \left(\mathbf{Diag}(p \circ v) \circ \mathbf{A}^\top \mathbf{A}\right) e = \sum_{i=1}^n p_i v_i \|a_i\|^2.
$$

To obtain (9), it remains to combine this with (13).

## C.5 Proof of Theorem 5

Similarly to the proof of Theorem 2, we use the update of Algorithm 1 to bound the following quantity

$$
\begin{aligned}
\mathrm{E}\left[\|x^{k+1} - x^\star\|^2 \mid x^k\right] &= \|x^k - x^\star\|^2 - \eta^k \sum_{i=1}^n \mathrm{E}\left[\left\langle \sum_{i \in S^k} \frac{1}{np_i} \mathcal{C}^k(g_i^k), x^k - x^\star \right\rangle \mid x^k\right] + \\
&\quad \mathrm{E}\left[\left\|\sum_{i \in S^k} \frac{\eta^k}{np_i} \mathcal{C}^k(g_i^k)\right\|^2 \mid x^k\right] \\
&\overset{(2)+(5)}{\leq} \|x^k - x^\star\|^2 - \eta^k \langle \nabla f(x^k), x^k - x^\star \rangle + \\
&\quad (\eta^k)^2 \left(\mathrm{E}\left[\left\|\sum_{i \in S^k} \frac{1}{np_i} \mathcal{C}^k(g_i^k) - \frac{1}{n}\sum_{i=1}^n \mathcal{C}^k(g_i^k)\right\|^2 \mid x^k\right] + \mathrm{E}\left[\left\|\frac{1}{n}\sum_{i=1}^n \mathcal{C}^k(g^k)\right\|^2 \mid x^k\right]\right) \\
&\overset{(2)+(5)+(9)}{\leq} \|x^k - x^\star\|^2 - \eta^k \langle \nabla f(x^k), x^k - x^\star \rangle + \\
&\quad \frac{(\eta^k)^2}{n^2} \mathrm{E}\left[\sum_{i=1}^n \left(\frac{\delta v_i}{p_i} + \delta - 1\right) \|g_i^k\| + \left\|\sum_{i=1}^n g_i^k\right\|^2 \mid x^k\right] \\
&\overset{(4)+(6)+(7)}{\leq} (1 - \mu\eta^k)\|x^k - x^\star\|^2 - 2\eta^k \left(1 - \eta^k \delta_{\mathbb{S}} L\right)(f(x^k) - f^\star) + \\
&\quad (\eta^k)^2 \left((\delta_{\mathbb{S}} - 1)D + (1 + a_{\mathbb{S}})\frac{\delta\sigma^2}{n}\right).
\end{aligned}
$$

Taking full expectation and $\eta^k \leq \frac{1}{2\delta_{\mathbb{S}} L}$, we obtain

$$
\mathrm{E}\left[\|x^{k+1} - x^\star\|^2\right] \leq (1 - \mu\eta^k)\mathrm{E}\left[\|x^k - x^\star\|^2\right] - \eta^k \mathrm{E}\left[f(x^k) - f^\star\right] + (\eta^k)^2 \left((\delta_{\mathbb{S}} - 1)D + (1 + a_{\mathbb{S}})\frac{\delta\sigma^2}{n}\right).
$$

The rest of the analysis is identical to the proof of Theorem 2 with only difference $c = (\delta_{\mathbb{S}} - 1)D + (1 + a_{\mathbb{S}})\frac{\delta\sigma^2}{n}$.

