# OpenReview forum: "A Better Alternative to Error Feedback for Communication-Efficient Distributed Learning"
_ICLR.cc/2021/Conference — ICLR 2021 Poster_

### Official Review · AnonReviewer2 · 2020-10-27
**I would recommend for now weak reject. I have some questions regarding the use of induced compressor and empirical experiments. I hope authors can address them in rebuttal.**

**Rating:** 5
**Confidence:** 3

**Review:**

Summary:

This paper proposes a framework of compressed communication that can be used to deal with error induced by contractive compressors and can serve as a superior alternative to the existing framework (compressed communication with error feedback (EF)). The proposed framework outperforms EF in terms of memory requirement, communication complexity, and technical assumptions. The extension to federated learning with partial participation, and the numerical experiments on real-world dataset illustrate the applicability of this framework.

Strength:

1. The paper compares two algorithms, i.e., distributed compressed SGD (DCSGD) and compressed communication with error feedback (EF), by analyzing their convergence rates. The paper provides a new theoretical analysis for DCSGD, with weaker technical assumptions and tighter bounds.

2. The induced compressor proposed in the paper can transform any biased compressor into an unbiased one, which can then be used in various existing methods designed for unbiased compressors.

Comments:

1. The paper concludes that the standard DCSGD algorithm is preferable to DCSGD with error feedback by comparing two algorithms' best-known convergence rates. It may not be appropriate because it is possible that the rate of standard DCSGD is theoretically tighter than that of DCSGD with error feedback (i.e., better theoretical bound). In other words, the difference in bounds of two algorithms is possibly due to the different theoretical analysis methods or different technical assumptions used in two algorithms, rather than the algorithm itself. How do those analysis methods used for finding convergence rates in two algorithms compared to each other? How can we make sure that the improvement is caused by the algorithm itself, but not the theoretical bounds?

2. Federated learning with partial participation has been studied in other works. I suggest the authors citing these papers and highlighting the differences. For example,
(1) Reisizadeh, Amirhossein, et al. "Fedpaq: A communication-efficient federated learning method with periodic averaging and quantization." International Conference on Artificial Intelligence and Statistics. 2020."
(2) Sattler, Felix, et al. "Robust and communication-efficient federated learning from non-iid data." IEEE transactions on neural networks and learning systems (2019).

3. The benefit of using a biased compressor is its low variance; this is what improves the performance. However, if a biased compressor is transformed into an unbiased one using an induced compressor, then don't we lose the benefits of using such a biased compressor? What is the benefit of transformation using an induced compressor if the variance is not low anymore? Why not use an unbiased compressor (e.g., C2) directly? Is it because of the additional unbiased "error feedback" the induced compressor can provide?

4. In experiments (Figure 3), authors compare various methods with one set of compressors (1) Rand-K; (2) Top-K + EF; (3) induced compressor with C1 = Top-K/2 and C2 = Rand-K/2; (4) Top-K, where K is a tunable parameter but is the same among all compressors. I wonder how their performances vary if the compressors use different values of K. For example, how do the compressor Rand-K/2 and the induced compressor with C1 = Top-K/2 and C2 = Rand-K/2 compare to each other (i.e., using unbiased C2 directly instead of transformation)?

---

> ### Author Response · Authors · 2020-11-22
> **We address all issues. Please reconsider the score.**
>
> Thank you for your very useful comments, suggestions and time.
>
> Issue 1
>
> Please note that our work has both a theoretical component and a practical component. In theoretical research, developing algorithms that prove better bounds is the  main goal of research. Whatever method has a better bound, however tight or not it is believed to be when compared against a practical run, is preferable, and methods with the best bounds are considered theoretical SOTA. Your comment ignores this very important point. The fact that our bounds are better is substantial. We request that you reevaluate your criticism from this perspective. Now, one can compare algorithms from a practical point of view, ignoring theory. Note that our main proposal: to use the induced compressor within DCSGD as opposed to the original contractive compressor + EF, is superior in practice as well, and often substantially so. So, what we have done offers the best from both worlds: better theory and better practical performance.
>
> Having said that, we strongly believe that the analysis of EF is tight, at least with respect to several parameter dependencies: we argue that the effect of averaging ($1/n$ factor which we get in our approach) can’t be obtained for EF. Please see our detailed response to AnonReviewer4 where we address this point in 4 paragraphs of text as a response tis his/her issue 1.
>
> Issue 2
>
> Thank you for listing these references, we will cite as appropriate and comment on the differences to our approach. The main distinctive feature is that in our analysis we have no assumption on the underlying distribution governing partial participation, whereas both these work assume (naive) uniform participation mechanisms.
>
> Issue 3
>
> While this is a valid observation, the situation is quite delicate. Compressing the error as we do in our induced compressor framework brings extra variance, but due to the unbiasedness of the induced compressor there is no need for EF in the pipeline, thus variance is reduced by a factor of $1/n$. Also, EF necessitates working with stale information, while we use all current information right away, albeit in a compressor form. While it is not clear a-priori what approach should be better, we show in theory and practice that our alternative to EF is better overall, often substantially so!
>
> As you suggest, using an unbiased compressor only is a very good baseline, and there are many methods and extensive research supporting unbiased compressors only. However, often biased contractive compressors are superior in practice, and until our work, EF was the only known mechanism that enables the use of biased compressors. So, the main goal of our work is not to argue that one should not use methods which utilize ${\cal C}_2$ only. Instead, our main message is: *if* for some application a biased compressor ${\cal C}_1$ offers superior performance to available unbiased alternatives, we suggest that one tries our induced compressor instead. Our construction is novel, simple and flexible, supported by theory and robustly superior across the experiments we have performed, a subset of which we included in the paper. Our approach benefits both from the greediness of ${\cal C}_1$ and from the unbiasedness of ${\cal C}_2$. Of course, ${\cal C}_2$ should be chosen wisely: it should not increase variance too much, and the per-round communication overhead should be controlled. We show (see Figure 3) that for our selection of ${\cal C}_2$, this is indeed the case. We plan to perform an extensive systems-level experimental evaluation in a follow-up work (currently underway) in which we will focus on the questions such as what is the right choice of ${\cal C}_1$ and ${\cal C}_2$ for specific tasks to provide guidelines for practitioners. These issues are delicate and necessitate additional work which is beyond the scope of this paper.
>
> If in a ${\cal C}_2$ only approach this compressor is not chosen wisely, then its performance is usually much worse in practice. For instance, standard random-K (picking K coordinates uniformly at random) does not compare well to clever Rand-K (Wangni et al.; see https://papers.nips.cc/paper/2018/file/3328bdf9a4b9504b9398284244fe97c2-Paper.pdf; Figures 1-4) or to Top-K with EF (see https://arxiv.org/abs/2002.12410 Figures 4-5).
>
> Issue 4
>
> Yes, it is fair to say that $K$ can be seen as a hyper-parameter. However, its value could also simply be imposed on practitioners by external circumstances such as limited bandwidth. We would like to stress here that in our experiments $K$ was not cherry picked in the sense that we only display such $K$ for which our method outperforms the baselines. We observed similar behavior along all the possible values of $K$.  Re proposed comparison with  Rand-$K/2$: please note that we already compare to a stronger baseline (Rand-$K$) which performs worse than our approach.

---

### Official Review · AnonReviewer1 · 2020-10-28
**Good paper, but experimental evaluation is very limited**

**Rating:** 6
**Confidence:** 3

**Review:**

This paper proposes an approach to reduce communication overhead in distributed training. The developed approach consists of a compressor operator (C(g)) that reduces the size of such messages (g). Specifically, the authors proposed a technique that transforms any biased compressor (e.g., Top-k) into an unbiased one, referred to as induced compressor, with better convergence guarantees and properties than the former. The advantages of the induced compressor are supported by a theoretical framework developed in the context of distributed compressed stochastic gradient descent (DCSGD). Moreover, the theoretical findings are validated with experiments in the CIFAR-10 dataset using two different image classifiers (Resnet-18 and VGG11 networks.).

Pros:
- The contribution of the paper is very nice. The idea to turn a biased compressor into an unbiased one is appealing and makes totally sense to me.
- The theory part is good and gives interesting insights.

Cons:
- The experimental evaluation is not convincing. It is restricted to one dataset only. Why did the authors not perform experiments on a second dataset to validate the approach?
- In my understanding one uses compression methods in FL to reduce the communication overhead. The proposed approach, however, seems to double the number of bits transmitted. The dimension "transferred bits" is missing in the experiments. I would like to see some comparison of Top-k + EF and the proposed approach wrt to bits transmitted vs. accuracy.
- Does the proposed approach also work well for non-iid data? How does the approach work with methods combining sparsity and communication delay, e.g., Sparse Ternary Compression (Sattler et al. 2020).

More experimental evaluation is required to better understand the benefits and limits of the proposed method.

---

> ### Author Response · Authors · 2020-11-20
> **Response: We will add more experiments, and the other issues are not issues at all.**
>
> Thank you for your valuable feedback, comments and time.
>
>
> Issue 1
>
> We include a small number of representative experimental results to make our point; and we believe that because we also have a theory to back our experimental results with, the point is made convincingly.
>
> Please note that our results are backed up by sound theory; and this means that one does not need to do as extensive experiments as one might need to do when designing a heuristic. Our theory predicts practical benefits well, as the experiments we included illustrate.
>
> Motivated by our theoretical and preliminary computational findings, we are already working on a follow-up paper focusing on an extensive experimental evaluation. In this work, we also focus on questions such as what is the right choice of ${\cal C}_1$ and ${\cal C}_2$ for a specific task to provide guidelines for practitioners. These questions are complex and require a dedicated and substantial piece of work.
>
> Issue 2
>
> Our approach does not need to double the number of communicated bits in each communication round. The example used in Section 3 is only one of many possibilities; and we included it for pedagogical reasons in order to illustrate one of the regimes in which our approach is expected to work well. Clearly, one may use a more aggressive compressor in place of ${\cal C}_2$, which comes at the expense of increasing the variance (see Theorem 3) and hence the number of communication rounds.
>
> Missing “transferred bit”: this is not actually an issue as all the methods we compared against each other communicate exactly the same number of bits in each round. We will make it more clear by adding an extra explanatory sentence.
>
> Issue 3
>
> Yes, our presented theory does *not* assume any relation among local data such as i.i.d. data, or gradient similarity among clients. This is clear from the formulation of our theorems. For the comparison part, we were, unfortunately, not able to find your provided reference as it was not provided completely. We would be happy to comment on this comparison if you could provide a link to the paper.

---

### Official Review · AnonReviewer4 · 2020-10-29
**General scheme for converting biased compressors to unbiased compressors - Not entirely convincing but adds to the discussion**

**Rating:** 7
**Confidence:** 4

**Review:**

**Quality and Clarity**

The paper is generally well written barring a few typos (see Queries and Suggestions below). The problem is clearly described and the solution is well motivated. The main theoretical result (Theorem 2) might be a bit hard to parse for readers unfamiliar with the theoretical results in this space but the overall explanation is clear.

**Originality and Significance**

The main approach for converting biased compressor to unbiased compressors appears to be novel and flexible. The theoretical analysis of distributed compressed SGD using unbiased compressors can also be useful to the community since the derived upper bound can be easily applied to any unbiased compressor.

**Strengths**

1. The approach for converting biased compressors to unbiased compressor does not seem to have been stated in this fashion before and provides an elegant framework for combining previous approaches.

2. The convergence analysis of unbiased compression is general and can be applied to any unbiased compressor and thus should be useful to the community as a whole.

3. The proposed approach appears to be more flexible than biased compression + EF and the authors show how it can be extended to Federated Learning settings, and also admit techniques like variance reduction and acceleration using momentum.

**Weaknesses**

1. Neither the theoretical analysis (comparison of upper bounds) nor the experiments (comparisons for some specific biased and unbiased compressors) in this paper convincingly show that the proposed approach will always be theoretically and practically better than Error Feedback.

2. The right choice of unbiased compressor $\mathcal{C}_2$ for a given biased compressor $\mathcal{C}_1$ is not clear. The authors show that if $\delta_2 = \delta_1$ then the upper bound of Theorem 2 will be tighter for the induced compressor $\mathcal{C}$ than for $\mathcal{C}_2$. However it is not clear if that will be enough to outperform EF since there will still be some extra variance. Indeed in the experiments the authors consider several unbiased compressors and while Top-a + Rand-(K-a) seems to beat Top-K + EF no general guidelines are offered.

2. The flexibility of the proposed approach which makes it amenable to the extensions  in Section 4 is not illustrated in the experiments which are limited to a direct comparison of the proposed approach with EF.

**Queries and Suggestions**

1. Why do both $\mathcal{C}_1(g)$ and $\mathcal{C}_2(e)$ need to be communicated if $\mathcal{C}$ is unbiased? (paragraph after Theorem 3)

2. While I understand that it is probably not possible to demonstrate that the proposed approach will outperform EF in every single scenario, I believe including some guidelines about choosing biased and unbiased compressors appropriately will greatly increase the impact of this work. For eg. Are all combinations of biased and unbiased compressors acceptable? Should one just care about ensuring that $\delta_1 = \delta_2$? What are good choices of unbiased compressors for some popular biased compressors? If it is possible to answer these questions, the paper will definitely be more useful to readers.

3. I would also recommend adding experiments on at least one of the extensions from Section 4 to highlight the flexibility of the proposed approach.

4. Typos:
i) Missing 'of' in Section 2, line 4.
ii) Last term in RHS of first equation in C.1 should be $+||x||^2$
iii) Missing '2L' in 3rd inequality in C.2
iv) Superscripts and subscripts have been interchanged in some terms in the proofs of Lemmas 6 and 7.
v) Second term in LHS of first equation of C.3 should be $\mathcal{C}_2(.)$

Overall, while I am not quite convinced that the paper achieves its stated goal of showing that the proposed unbiased compression approach always outperforms biased compression + EF, I believe it adds enough to the discussion in this space to merit acceptance.

**Comments after Author Response**

I thank the authors for their response. My opinion of the potential of this paper to encourage further discussion in this area is unchanged. I can already see from the response on choice of biased and unbiased compressors to combine that there is plenty of scope for future work that builds on this idea. Regarding the comparison with EF, I appreciate the additional intuition provided in the author response on the drawbacks of EF and hope to see improvements to EF or more exhaustive comparisons between EF and the approach proposed in this paper in future work. As I had already recommended acceptance I am leaving my score unchanged.

---

> ### Author Response · Authors · 2020-11-20
> **Response to all 3 weaknesses: 1) is not an issue; we will add some guidelines to address 2); we will do experiments to address 3)**
>
> Thank you for your valuable comments, positive feedback and time!
>
> Weaknesses:
>
> 1. We respectfully disagree. Our theory does provide better bounds than EF, as we argue in the paper, and our experiments do clearly show benefits in practice in comparison to EF. We note you did not provide any justification for your claims. We have conducted more experiments than the ones we included in the paper and the results we included are presentative.
>
> Moreover, there are many reasons to believe that existing analysis of EF is tight with respect to the parameter dependencies. For instance, we believe that the effect of averaging ($1/n$ factor) can’t be obtained for EF without (not without further strong assumptions). Indeed, intuitively, EF can't benefit from averaging since every client has a different error relevant to its own past communication rounds, and all these errors need to be corrected. Since $\frac{1}{n} \sum_{i=1}^n error_i = error$, having more devices does not limit the “global” error, which is of the same order as local errors. This is also what existing EF bounds show: indeed, this is consistent with the most recent SOTA convergence guarantees of Gorbunov et al (https://arxiv.org/pdf/2010.12292.pdf,  NeurIPS 2020) who also obtain a rate without the $1/n$ factor.
>
> On the other hand, if each device communicates an unbiased estimate of the true update, the only price we pay is the extra variance, which decreases with increasing number of clients due to averaging (let $X_i$ be a zero mean distribution with variance $\beta$, then average of $X_i$’s is a zero mean distribution with variance $\frac{\beta}{n}$).
>
> From a practical point of view, Figure 1 illustrates the necessity of some sort of EF in order to guarantee convergence. We show that a simple unbiased alternative performs significantly better.  We also refer you to our ablation study in Figure 2, where we compare such setting that the only difference in 2 compared methods is that one algorithm uses EF and the other one does not. It can be seen that the gap is significant, even for reasonably small $n=8$. This matches our theoretical results. For Figure 3, we can see that EF performs well, which is not because of EF itself, but mainly due to the low empirical variance. It can be seen that our simple construction of induced compressor performs better as it benefits both from the greediness of ${\cal C}_1$ and from the averaging effect due to unbiasedness. This agrees with our theory. We will expand our discussion to make these points even more clear.
>
> 2. You are right. We will include some guidelines along these lines in the paper. However, a full treatment of this question is beyond the scope of our paper - see our response #1 to AnonReviewer3 who asked the same question.
>
> 3. Good suggestion. We will include additional experiments to selected extensions disused in Section 4, as requested. We did not do it before due to space limitation and because we believed the message is already clear and convincing. However, we agree additional experiments of this type will be beneficial.
>
>
> Queries and suggestions: thank you, these are all very helpful!
>
> 1.  Of course, only the final output of ${\cal C}$ needs to be communicated. However, one obvious and appealing way of doing that is to communicate both ${\cal C}_1(q)$ and ${\cal C}_2(e)$, and combine them at the master node. While in special cases it may be possible to do better than this (we did not explore this), this remains our default suggestion for how $\cal C$ could be communicated effectively. We will make this more clear in the paper.
>
> 2. This is very relevant to our second answer in the weaknesses part. For the $\delta_1 = \delta_2$ suggestion, we don’t argue that this is optimal or a good choice, we just use it for illustration. In theory, all combinations of biased and unbiased compressors will work, but there is price to be paid: some choices will lead to better (faster) methods than others.
>
> 3. As mentioned in our response to Weakness 3, we will add more experiments. Thanks for the suggestion!
>
> 4. Thank you for spotting these typos, we will fix them!

---

### Official Review · AnonReviewer3 · 2020-11-02
**Recommendation to Accept**

**Rating:** 9
**Confidence:** 4

**Review:**

Summary: The paper studies compression of gradients in distributed training with the goal of minimizing communication costs.  The main contribution of the paper is the development of an *induced compressor* that takes as input, a possibly biased compressor, and outputs an unbiased compressor with similar error variance as compared with the original compressor.  As the output of the induced compressor is unbiased, it obviates the need for error feedback, which is needed when biased gradient estimates are used.

Reason for Score:

I vote for accepting the paper.
--> The idea of an induced compressor is simple (after the fact), and elegant, and advances state of the art in gradient compression.
-->The paper has solid theoretical reasoning via a convergence analysis that does demonstrate that for certain (common) parameter choices the induced compressor is expected to have better convergence than the biased counter part
-->   Satisfactory experiments to back up the theoretical insights. In particular, the output of the induced compressor requires more bits as compared with the underlying biased compressor. Empirical results show that, an induced compressor obtained from a class of biased compressors (e.g., top K gradients) has better performance than a biased compressor from that class with the same communication cost.


Pros:

--> This paper resolves a problem of current gradient compression techniques: the good compressors that require a small number of bits/values to represent the gradients are inevitably biased, and the bias needs to be compensated via error feedback techniques. It shows that there are good unbiased compressors and provides a recipe for constructing them.

--> The paper is well-written and based on a sound theoretical reasoning.

Cons:
--> I would like a more detailed discussion on the effect of the unbiased compressor, that is used in the development of the induced compressor. How sensitive is the performance/compression to the choice of the unbiased compressor?
--> I wonder how the theory would apply to weakly convex or non-convex settings.

==================================
Comment after author responses.
On reading the author's responses to my (and other) review comments, my recommendation remains unchanged. This is a solid piece of work.

---

> ### Author Response · Authors · 2020-11-20
> **Effect of the inducing compressor and extension to weakly convex and nonconvex settings**
>
> Thank you for your valuable comments, very positive feedback and time. We share your enthusiasm.
>
> 1. Choice of the unbiased compressor ${\cal C}_2$ used to define/generate the induced compressor ${\cal C}$ (i.e., choice of the "inducing" compressor).
>
> Indeed, the question of the appropriate choice of the inducing compressor ${\cal C}_2$ is important. However, it is also not trivial, and our preliminary studies show that there is much to be discovered here, both in theory and in terms of developing further practical guidelines to those already contained in our paper.
>
> Note that Theorem 3 gives a formula for $\delta$ parameter of the induced compressor ${\cal C}\in \mathbb{U}(\delta)$. The larger this parameter is, the more iterations will a gradient type method using our induced compressor $\cal C$ need to perform (to achieve the same guarantee). This is true for all distributed gradient-type methods using communication compression, including DCSGD (Algorithm 1), DIANA (=variance reduced version of DCSGD), and ADIANA (accelerated version of DIANA). Since by Theorem 3 we have $\delta = \delta_2 \left(1- \frac{1}{\delta_1}\right) + \frac{1}{\delta_1}$, it is clear that $\delta$ increases linearly with $\delta_2$ (the variance parameter of ${\cal C}_2$). So, in order to keep $\delta$ small, we need to keep $\delta_2$ small. However, small $\delta_2$ also means that ${\cal C}_2$ offers less compression, which means that more bits are communicated in each round. So, one needs to strike a balance. Intuitively, it makes sense to choose ${\cal C}_2$ so that i) it transmits no more than the number of bits transmitted by ${\cal C}_1$ (in this way, the number of communicated bits in each communication round at most doubles), and ii) so that $\delta_2$ is as small as possible. This will in general lead to a complicated optimization problem over al unbiased compressors with a constraint on the number of bits.
>
> Moreover, notice that, for instance, Rand-$K$ and induced compressor with ${\cal C}_1$ set to Top-$a$ and ${\cal C}_2$ set to Rand-$(K-a)$ (providing fixed budget on bits) are both unbiased and they give the same worst-case bound on the variance. Hence, optimizing over $a$ in theory does not lead far. A big portion of the practical benefits of our induced compressor can be attributed to i) the adaptive/greedy nature of ${\cal C}_1$ (chosen as Top-$a$, which has low empirical variance) along the path of the iterates, and not from the worst case behavior captured by theory, and ii) the variance reduction effect independent unbiased compressors offer when the number of workers $n$ is large (this effect is not present in EF).
>
> In summary, the question of (theoretically) optimizing for ${\cal C}_2$ is difficult, as it necessitates a deeper theoretical understanding of biased compressors, which is currently missing. An alternative is to impose some assumptions on the structure of gradients encountered during the iterative process, or to perform an extensive experimental evaluation on desired tasks to provide guidelines for practitioners. We are currently looking into all of these questions; but consider them beyond the scope of this paper. We plan to release our findings as a follow-up to this work.
>
> 2. Application of theory to convex and non-convex functions
>
> First, please note that we only require $\mu$-quasi strong convexity, which is a condition that does not imply convexity. Second, our theory can indeed be extended to cover general convex or non-convex objectives, and unlike EF, our induced compressor would still benefit from averaging ($\delta_n$ and $\sigma^2/n$ terms). Thus, the story is essentially the same. Lastly, note that since our induced compressor belongs to $\mathbb{U}(\delta)$, and since several distributed first order methods already support compressors from this class and have been analyzed in the convex and nonconvex regimes, our induced compressor clearly has immediate applicability in these regimes as well.

---

### Decision · Program_Chairs · 2021-01-07
**Final Decision**

**Decision:**

Accept (Poster)

**Comment:**

I agree with the reviewers' comments. The technique proposed in the paper is very interesting, and although the method itself is not particularly surprising (it's "just" chaining two compressors), it's a really nice way of framing and studying the problem. On the other hand, the experiments _are_ relatively weak, and I think there is significant potential for improvement here (especially with an added 9th page of text). I encourage the authors to add some more convincing experiments in future versions of the paper.